# Self-Reported Physical Fitness in Children and Adolescents with Obesity: A Cross-Sectional Analysis on the Level of Alignment with Multiple Adiposity Indexes

**DOI:** 10.3390/children8060476

**Published:** 2021-06-04

**Authors:** Matteo Vandoni, Nicola Lovecchio, Vittoria Carnevale Pellino, Roberto Codella, Valentina Fabiano, Virginia Rossi, Gian Vincenzo Zuccotti, Valeria Calcaterra

**Affiliations:** 1Laboratory of Adapted Motor Activity (LAMA), Department of Public Health, Experimental Medicine and Forensic Science, University of Pavia, 27100 Pavia, Italy; vittoria.carnevalepellino@unipv.it; 2Department of Human and Social Science, University of Bergamo, 24127 Bergamo, Italy; Nicola.lovecchio@unipv.it; 3Department of Industrial Engineering, University of Rome Tor Vergata, 00133 Rome, Italy; 4Department of Biomedical Science for Health, University of Milan, 20133 Milan, Italy; roberto.codella@unimi.it; 5Department of Endocrinology, Nutrition and Metabolic Diseases, IRCCS MultiMedica, 20122 Milan, Italy; 6Department of Biomedical and Clinical Science “L. Sacco”, University of Milan, 20157 Milan, Italy; valentina.fabiano@unimi.it (V.F.); virginia.rossi@unimi.it (V.R.); gianvincenzo.zuccotti@unimi.it (G.V.Z.); 7Pediatric Department, “Vittore Buzzi” Children’s Hospital, 20154 Milan, Italy; valeria.calcaterra@unipv.it; 8Pediatric and Adolescent Unit, Department of Internal Medicine, University of Pavia, 27100 Pavia, Italy

**Keywords:** children with obesity, self-reported physical fitness, physical activity, adiposity indexes

## Abstract

Obesity has been associated with several alterations that could limit physical activity (PA) practice. In pediatrics, some studies have highlighted the importance of enjoyment as a motivation to begin and maintain adherence in PA. Since self-reported physical (SRPF) fitness was related to motivation, the aim of this study was to investigate the existence of differences between SRPF in children with obesity (OB) compared to normal weight (NW). The International Fitness Enjoyment Scale (IFIS) questionnaire was administered to 200 OB and 200 NW children. In all the subjects, height, weight, and BMI and in OB children adiposity indexes including waist circumference (WC), body shape index (ABSI), triponderal mass index (TMI), and fat mass were measured. NW group showed higher IFIS item scores than the OB group (*p* < 0.01), except in muscular strength. In OB, the anthropometric outcomes were inversely correlated to SRPF outcome except for muscular strength. OB children reported a lower perception of fitness that could limit participation in PA/exercise programs. The evaluation of anthropometric patterns may be useful to prescribe a tailored exercise program considering individual better self-perception outcomes to obtain an optimal PA adherence.

## 1. Introduction

Obesity is defined as a condition characterized by an excessive fat accumulation that has negative health consequences [1,2]. Childhood obesity is a grave public health concern with increasing prevalence all over the world with long-term medical, social, and economic consequences [1]. According to the World Health Organization report, the global prevalence of overweight and obesity in children and adolescents aged 5–19 has risen from 4% in 1975 to 18% in 2016 [2]. In 2016, more than 340 million children and adolescents worldwide were in a condition of excess body weight [2]. The excess of weight during childhood caused by sedentary habits and unhealthy lifestyle led to increased risk of metabolic and cardiovascular disorders such as dyslipidemia, hypertension, and insulin resistance, hallmarks of metabolic syndrome (MetS), more frequently during adolescence [3,4]. Moreover, obesity has been associated with several alterations that could limit physical activity (PA) practice, with a consequent worsening of the quality of life. In fact, compared to normal weight peers, children with obesity tend to have lower PA levels, thus increasing the self-perceived barriers to sport and PA participation. Altogether, these factors may lead to direct repercussions on adipose tissue gain, augmented cardiovascular disease (CVD) risk into adulthood, and limitations in social–peer relationships [5,6].

In pediatrics, some studies have highlighted the importance of enjoyment in increasing intrinsic motivation (perceived autonomy, success, and competence) to begin and maintain adherence in PA [7]. Motivation is a complex human factor that distinguishes the reason for a specific action [8,9]. In particular, behavioral regulation toward an activity can be intrinsically motivated (self-determined), extrinsically motivated (controlled), or amotivated (non-intentional) [10]. Intrinsic motivation reflects situations in which individuals perform an activity to experience fun, learn new things, or develop their competence. In contrast, extrinsic motivation is represented in situations in which an individual performs activities with desirable outcomes in mind. The intrinsic motivation theory provides a model that motivates the nature of enjoyable experiences that could enhance positive attitudes toward and encourage PA participation [9,11]. Then, the high level of self-reported physical fitness (SRPF) was founded as a precursor of enjoyment [12,13]. In general, SRPF refers to individuals’ perception of their actual physical fitness. Ortega et al. [14] previously reported that a lower SRPF is associated with a worst cardiovascular profile and risk of weight gain in children and adolescents. Moreover, Stodden et al. [15] proposed a model indicating SRPF as a strong carrier for promoting positive adherence to PA. In particular, an inverse relationship was found between anthropometric characteristics and SRPF, in which children with higher BMI had a lower SRPF and motivation during physical exercise compared to their normal-weight peers [16,17]. These conditions could contribute to developing a negative spiral of disengagement in PA with low SRPF, less PA level, and poor health-related physical fitness. In childhood, motivation to PA practice and a higher level of enjoyment seem not to be influenced by gender, instead parental choices and family support play a key role in real PA participation [16]. Whether SRPF outcomes, except for flexibility, in normal weight boys seems to be higher than girls at this age [18], gender differences in SRPF of boys and girls with obesity seems to remain unclear. To this end, sports specialists and pediatricians could ameliorate and tailor PA proposals for children with obesity. In fact, since girls and boys usually tend to have different inclinations in PA and sport choice, a better self-confidence in an outcome could highlight some starting points to differentiate sport and exercise prescription by gender.

Even if BMI is mainly used in the evaluation of anthropometric characteristics [19], the introduction of more precise indexes could assist in the evaluation of body composition and cardiometabolic risk. Waist circumference (WC) reflects fat distribution and fat percentage [20] and it is considered as a predictor of hypertension and impaired glucose metabolism when compared to BMI [21]. Although WC is a better marker of abdominal fat accumulation than BMI, an elevated waistline alone is not sufficient to diagnose visceral obesity and therefore the MetS risk. Recently, a new index, the body shape index (ABSI), related to the abdominal to peripheral fat ratio, has been specifically developed to stress the importance of WC in abdominal obesity, associated with metabolic and cardiovascular alterations [22,23,24]. Additionally, triponderal mass index (TMI) has been recently suggested as a useful tool in the body composition evaluation [25] and has been studied as predictors of MetS [26] in pediatrics. Finally, it is well known that BMI does not discriminate between lean and fat-free mass, therefore the non-linear equation developed by Hudda et al. [27] has emerged as more effective as it estimates the fat mass with weight, height, ethnic origins, and age.

To the best of our knowledge, there is a lack of data on SRPF in obese children that could help understand possible self-barriers in PA practice and sport adherence. The study of SRPF and its relationship with recent anthropometric adiposity indexes may be insightful as a primary health care strategy. As a secondary endpoint, PA participation and enjoyment, which are crucial intrinsic motivation, are also a focus.

The aim of this study was to investigate the differences between SRPF in children with obesity compared to normal weight subjects along with the relationship between SRPF and anthropometric adiposity indexes. 

## 2. Materials and Methods

### 2.1. Study Design and Participants

We conducted a cross-sectional study on a total of 200 Caucasian children (*n* = 85 females, 10.2 ± 1.5 years) with obesity (BMI-z score ≥2 according to World Health Organization), aged 10.2 ± 1.5 years (range 8–12 years) were enrolled from Pediatric Hospital Vittore Buzzi Children’s Hospital of Milan (Italy). Subjects were referred to our institution for obesity by their general practitioner or by their primary care pediatric consultant. 

The patients were asked to participate in the study during a 50-min pediatric specialistic visit and an International Fitness Enjoyment Scale (IFIS) questionnaire was administered by both pediatricians and sports specialists. 

Inclusion criteria were aged between 7 and 12 years, BMI-z score <1 for normal-weight children and ≥2 for children with obesity and the Italian language proficiency. Exclusion criteria were known secondary obesity conditions, non-comprehension of Italian language, cardiovascular and respiratory chronic diseases, comorbidities, orthopedic injuries, and absolute contraindications to the PA practice. 

As control groups for IFIS items, 200 normal-weight (BMI-z score <1, according to the World Health Organization) subjects, comparable for age and sex (*n* = 101 females, 9.6 ± 1.3 years), admitted for auxological evaluation, were considered. 

The institutional ethics committee approved the study (protocol numbers 2020/ST/234, 2020/ST/298) and it was conducted in accordance with the Helsinki Declaration of 1975, as revised in 2013 [28].

### 2.2. Measures

#### 2.2.1. Anthropometric Characteristics and Adiposity Indices

In all the subjects, height, weight, and BMI were measured. Weight was quantified with participants not wearing shoes and in light clothing, standing upright in the center of the scale platform (Seca, Hamburg, Germmany) facing the recorder, hands at sides, and looking straight ahead [29].

Standing height was measured using a Harpenden stadiometer (Holtain Ltd., Crosswell, UK) with a fixed vertical backboard and an adjustable head piece. The measurement was taken on the child in an upright position, without shoes, with their heels together and toes apart, hands at sides, aligning the head in the Frankfort horizontal plane [30]. Two measurements were taken for each parameter, and a third was obtained if a discrepancy was noted between the initial measurements for weight (>500 g) and height (>0.5 cm). Final growth parameter values were based on the average of the two closest measurements. BMI was calculated as body weight (kilograms) divided by height (meters squared) and was transformed into BMI z scores using WHO references [20]. 

In children with obesity, adiposity indexes including WC, ABSI, TMI, and fat mass were also calculated as follows: WC measured in a standing position, at the end of a quiet expiration, at midpoint between costal margin and iliac crest [30];ABSI = 1000*WC*Wt ^–2/3^*Ht^5/6^ [31];TMI = weight (kg)/ height (m)^3^ [31]; andFat Mass = eight − exp[0.3073 × height^2^ − 10.0155 × weight^−1^ + 0.004571 × weight − 0.9180 × ln(age) + 0.6488×age^0.5^ + 0.04723×male + 2.8055] [27]

(exp = exponential function, ln = natural logarithmic transformation, male = 1, female = 0)

#### 2.2.2. International Fitness Enjoyment Scale (IFIS)

IFIS is a self-reported, simple, and short fitness scale previously validated in nine European countries and languages that defined physical fitness as an indicator of physical competence [14,32]. The IFIS consists of a 5 point Likert scale (from 1 very poor to 5 very good) with questions focused on five macro-areas of fitness: general fitness, cardiorespiratory, strength, speed-agility, and flexibility. The IFIS had high validity and moderate-to-good reliability (average weighted Kappa: 0.70 and 0.59) for aged-school children. The questionnaire was administered during a specialist visit by the same sports science expert.

### 2.3. Statistical Analysis

All quantitative data were summarized as mean (SD) or median (range) as appropriate. We tested for normality by Shapiro–Wilk tests and graphically checked for linearity. We used a non-parametric two-way ANOVA Friedman rank test to evaluate the differences between groups, normal weight (NW), and children with obesity (OB) and gender as independent variables and each item of the IFIS questionnaire as dependent variables. The Bonferroni test was eventually used as post-hoc analysis. Then, only in the OB group, we provided a Spearman rho correlation [33] to determine the possible relationship between anthropometric adiposity indexes and each item of IFIS. All the significance was set at a *p*-value less than 0.05. Regarding the sample size, with a power of 0.90 and an alpha error of 0.05, we conveniently enrolled 200 children for each group [34]. Statistical analyses were performed using “The Jamovi project (2021). *Jamovi* Version 1.6 for Mac [Computer Software], Sydney, Australia; retrieved from https://www.jamovi.org.

## 3. Results

The anthropometric data in OB and NW subjects are shown in Table 1. A significative difference (*p* < 0.001 between OB and NW groups were noted in height (*p* = 0.000), weight (*p* = 0.000), BMI *p* = 0.000), and BMI Z-score (*p* = 0.000), both in males and females.

In Table 2, adiposity indexes in the OB group according to sex are reported. No significant differences were noted between males and females (*p* > 0.05).

Between-group differences for each IFIS item (general physical fitness, cardiorespiratory fitness, speed/agility, and flexibility) are reported in Table 3. Particularly, the NW group showed higher IFIS item scores than the OB group, except in muscular strength for which the OB group presents the highest value (*p* < 0.05). These differences persist by showing the same pattern when results were stratified by gender. In general, males showed higher values than females in all the IFIS items (*p* < 0.05), except for flexibility. We also found that males in the NW group showed higher values than males in the OB group for physical fitness, cardiorespiratory fitness, and speed/agility, while females in the NW group showed higher values than females in the OB group for flexibility. No significant differences were found for muscular strength between males in the NW and OB groups. Finally, the OB female group showed higher muscular strength values compared to males in the OB and NW groups, and females in the NW group. 

Two-way ANOVA analysis revealed that OB and NW groups had significant differences in all the IFIS items. By comparing gender, we found that only flexibility was higher in females confirmed by the Bonferroni post-hoc test.

In Table 4, the correlations between anthropometric adiposity indexes and IFIS items are reported. Fat mass was inversely and significantly correlated (*p* < 0.01) with all IFIS items, except for muscular strength. WC showed a significant level of negative correlation in perceived general physical fitness and muscular strength (*p* < 0.05). The ABSI was shown to be significantly and positively correlated with perceived cardiorespiratory fitness and speed/agility (*p* < 0.01).

## 4. Discussion

Despite clear benefits demonstrated from practicing regular PA as well as the fact that a healthy lifestyle can reduce cardio-metabolic risk factors, children with obesity tend to have lower PA levels [5,6,35]. Our study aimed to investigate the differences between SRPF in children with obesity compared to normal weight children. We found a difference (lower perception in children with obesity) for all the outcomes except for the muscular strength item. Stodden’s model underlined the importance of SRPF for implementing PA practice, suggesting an inverse relationship between BMI status and SRPF [15,17,36]. In this study, we investigated the existence of differences between SRPF in children with obesity compared to normal weight subjects and we explored the extent of alignment between SRPF and various anthropometric adiposity indexes. Our results showed that the OB group tended to have lower SRPF than the NW group except for muscular strength perception. Additionally, in both groups, females tended to have lower SRPF compared to males except for flexibility. These results are in accordance with previous studies provided by Ortega et al. [14] and Lovecchio et al. [36], where children with lower BMI had higher SRPF. In fact, low physical self-efficacy could be due to excessive weight, and females were found to be worst affected by negative body-image and social pressure [37,38]. Surprisingly, obese females had higher values of muscular strength perception with respect to their male peers. This result is novel with regard to SRPF and could be determinant for increasing PA adherence in youth, especially in overweight/obese girls. In fact, a low SRPF is a crucial point to address because it represents a clear self-barrier for obesity when approaching a regular exercise program. Babic et al. [39] showed that adolescents with a higher level of physical self-esteem had more chances to engage in PA. Conversely, lower perception of fitness could limit participation in PA/exercise programs and, consequently, the enjoyment perceived [15,35]. Moreover, in children with obesity, psychophysical impairment related to motor skill difficulties and low self-esteem may reduce PA adherence, creating a vicious circle with direct repercussions on physical fitness, self-esteem gain, and greater CVD risks. An accurate evaluation of SRPF, particularly in youths, could identify low-fit (and possibly low-active) children to enact primary surveillance and design specific PA interventions. We emphasized the need to scrutinize SRPF in children with obesity to engage them with an efficiently adapted and attractive PA program. Given a greater perception of muscular strength in obese females, one could promote this kind of exercise stimulation as an early approach in overweight/obese children.

A preliminary and accurate evaluation of anthropometric patterns in developmental age could ameliorate the individualized exercise program prescription, guaranteeing PA adherence and a more active lifestyle in children with obesity [15]. In particular, not only BMI, but also ABSI and fat mass outcomes seemed to be better predictors of higher SRPF, and subsequently could enhance a greater PA and exercise adherence [27,31]. In fact, we observed SRPF scores decreasing as the anthropometrics adiposity indexes such as ABSI and fat mass were increasing, especially in the general physical fitness and cardiorespiratory fitness domains. The pattern varied when considering multiple indexes. In detail, all the anthropometric indexes revealed a decrease in all the SRPF dimensions except in muscular strength. These results underlined the necessity to better evaluate body composition upon commencing a PA program, ideally tailored on adapted requirements. Children with obesity tend to have negative feelings related to their cardiorespiratory fitness domain due to greater difficulty in performing motor skills (primarily due to increased overall mass) [40]. This is also true with regard to multiple fitness capacities, but not for muscular strength perception. For these reasons, to adequately meet the obese children’s compliance, we advocate that pediatrics and sports specialists may personalize the PA program, suggesting activities and motor tasks that produce positive feelings like increased self-perception and self-esteem (i.e., strength activities) [41]. Finally, in accordance with the anthropometric adiposity indexes used, we propose to start a PA program combining diverse activities such as speed agility, strength, or general performance and, stepwise, moving forward to enhance PA confidence, the self-promotion of an active lifestyle and full PA adherence. 

Since this study investigated SRFPF and not a real performance, we are unable to objectively evaluate the correctness of children’s perception. On the other hand, SRFPF is easy to evaluate and gives a key information on children’s predisposition to exercise, highlighting possible lacks or sense of inadequacy in some components of physical fitness. Moreover, this study did not show any data on educational and cultural status of children that could improve the understanding of SRPF. Future studies should focus on comparing real children’s performance with the new adiposity indexes and investigate the relationship of parental-socioeconomic status and SRPF. Finally, the comparison of SRPF with motivation outcomes could enhance knowledge on this topic.

## 5. Conclusions

In conclusion, in our study, children with obesity showed a lower SRPF compared to normal weight subjects, except in muscular strength perception. Preliminary results showed a correlation between anthropometric adiposity indexes and IFIS items were detected, suggesting that the evaluation of anthropometric patterns could help the prescription of a tailored exercise program. A variety of different activities should be considered to improve PA adherence.

## Figures and Tables

**Table 1 children-08-00476-t001:** Anthropometric characteristics of the whole sample.

	Total NW	Total OB	Males NW	Males OB	Females NW	Females OB
Age (years)	9.59 ± 1.31	10.20 ± 1.46	9.37 ± 1.10	10.30 ± 1.46	9.79 ± 1.47	10.20 ± 1.47
Weight (kg)	30.52 ± 3.95	51.10 ± 12.40 *	30.44 ± 3.73	52.70 ± 12.50 *	30.59 ± 4.17	48.80 ± 12.00 *
Height (cm)	136 ± 7	142 ± 11 *	136 ± 8	144 ± 10 *	135 ± 7	139 ± 11 *
BMI (Kg/m^2^)	16.47 ± 0.87	24.90 ± 3.18 *	16.35 ± 0.72	25.00 ± 3.19 *	16.58 ± 0.98	24.7 ± 3.19 *
Z-Score	−0.05 ± 0.42	1.80 ± 0.33 *	0.01 ± 0.44	1.84 ± 0.31 *	−0.09 ± 0.41	1.75 ± 0.35 *

All the data are shown as mean ± DS. Legend: BMI = Body Mass Index; WC = Waist Circumference; ABSI = A Body Shape Index; TMI = Tri-Ponderal Mass Index; NW = Normal Weight; OB = Obese. * *p* < 0.05.

**Table 2 children-08-00476-t002:** Anthropometric characteristics of the children with obesity (OB).

	Total OB	Males OB	Females OB
WC (cm)	88 ± 10	89 ± 10	86 ± 10
ABSI	0.09 ± 0.01	0.09 ± 0.01	0.09 ± 0.01
TMI (kg/m^3^)	17.50 ± 2.06	17.30 ± 2.00	17.80 ± 2.12
Fat mass (Kg)	19.60 ± 5.95	19.70 ± 5.77	19.60 ± 6.22

All the data are shown as mean ± DS. Legend: BMI = Body Mass Index; WC = Waist Circumference; ABSI = A Body Shape Index; TMI = Tri-Ponderal Mass Index; NW = Normal Weight; OB = Obese.

**Table 3 children-08-00476-t003:** Differences for each International Fitness Enjoyment Scale item.

	Total NW	Total OB	*p*-Value	Males NW	Males OB	*p*-Value	Females NW	Females OB	*p*-Value
General physical fitness	4.26 ± 0.76	3.54 ± 1.12	<0.001	4.29 ± 0.76	3.54 ± 1.16	<0.001	4.22 ± 0.77	3.53 ± 1.08	<0.001
Cardiorespiratory fitness	4.01 ± 0.95	3.35 ± 1.07	<0.001	4.11 ± 0.92	3.37 ± 1.13	<0.001	3.90 ± 0.96	3.33 ± 0.98	<0.001
Muscular strength	3.98 ± 0.96	4.18 ± 0.84	0.013	4.09 ± 0.91	4.10 ± 0.86	0.843	3.85 ± 0.99	4.31 ± 0.80	<0.001
Speed/agility	4.30 ± 0.77	3.48 ± 1.13	<0.001	4.39 ± 0.75	3.43 ± 1.16	<0.001	4.21 ± 0.79	3.43 ± 1.07	<0.001
Flexibility	3.85 ± 0.99	3.25 ± 1.13	<0.001	3.70 ± 1.05	3.13 ± 1.05	<0.001	4.0 ± 0.92	3.42 ± 1.22	<0.001

All values are shown mean ± DS. Legend: NW = Normal Weight; OB = Obese.

**Table 4 children-08-00476-t004:** Spearman rho correlations between the anthropometric indexes and International Fitness Enjoyment Scale in children with obesity.

	General Physical Fitness	Cardiorespiratory Fitness	Muscular Strength	Speed/Agility	Flexibility
WC	−0.182 *	−0.012	−0.175 **	−0.062	−0.092
BMI	−0.293 ***	−0.188 **	0.016	−0.242 ***	−0.189 **
ABSI	0.071	0.198 **	−0.135	0.208 **	0.045
TMI	−0.208 **	−0.169 *	0.094	−0.131	−0.012
Fat Mass	−0.286 ***	−0.206 **	0.019	−0.304 ***	−0.211 **

* *p* < 0.05; ** *p* < 0.01; *** *p* < 0.001. Legend: WC = Waist Circumference; BMI = Body Mass Index; ABSI = A Body Shape Index; TMI = Tri-Ponderal Mass Index; NW = Normal Weight; OB = Obese.

## Data Availability

The data presented in this study are available on request from the corresponding author. The data are not publicly available due to privacy reasons.

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
