# Peer review of "Self-Reported Physical Fitness in Children and Adolescents with Obesity: A Cross-Sectional Analysis on the Level of Alignment with Multiple Adiposity Indexes"

_children, 2021, doi:10.3390/children8060476_

Round 1

Reviewer 1 Report

This study aims to analyse and investigate the existence of differences between the SRPF in children with obesity compared to subjects of normal weight and to investigate the relationship between the SRPF and the different anthropometric indices of adiposity. 

I make the following recommendations below:

Method: 

Unify the sample description criteria, line 96 in obese sample and line 108 normal weight sample.

What were the instruments used to measure height and weight and their accuracy?

Why was the IFIS questionnaire chosen and not another existing one?

I would like to know what is new about this study and what it contributes to this line of research. The results obtained are obvious and frequently studied. 

Reviewer 2 Report

General comment:

First of all, I would like to say that I am very thankful to have the opportunity to read this study. The suggestions given in this document are intended to improve your work. If you do not agree with any of them, please explain them to me, and we will try to reach a consensus. The same feedback document will be given to both editors and authors.

I think this article is relevant, and useful for health professionals, but there are issues that I think should be addressed:

Introduction section:

  • Although the introduction is well written in general terms, more information should be given on the key aspects addressed in the manuscript. For example, Childhood obesity (definition, prevalence, consequences...); Motivation is a very broad concept that should be further defined; In line 57 it is stated that “In pediatrics, some studies highlighted the importance of enjoyment as a motivation to begin and maintaining adherence in PA” how?

  • The aim is written as if it were a hypothesis. Check this, please.

    Methods section:

  • Section 2.1 Study design and participants does not include information on the study design, but it does include information on the procedure and ethical aspects. Please rewrite this part.

  • Do the authors have no sociodemographic data other than age and sex? It would be very useful to characterize the sample better.

  • Please, include inclusion criteria.

  • Please check for missing references. For example, line 112.

  • I think it would be advisable to report the statistical power of the study.

    Results section:

  • Authors present the differences between boys and girls in the results, but this aspect is not accurately reflected in the previous sections of the paper. Please review this aspect and indicate why gender could be an important factor.

Discussion and conclusion section:

  • I think the limitations and future lines should be expanded.

  • Perhaps the authors should consider making the conclusion more conservative based on the power of their study.

Round 2

Reviewer 1 Report

The authors have improved the manuscript and the paper can be accepted.

There is no a .000 p value, the authors should change this information in the table and include >0.001

Author Response

Thanks for the note, we corrected it. We appreciated your revision that improved our manuscript

Reviewer 2 Report

I think that the quality of the manuscript has increased. There are some of my impressions:

  • The introduction and the aims are now clearer.
  • I think that Ethical concerns should have an independent subsection.
  • References about the instrumentation used should be provided.
  • Table 1 has a different format.
  • I think the discussion can be improved with a few simple adjustments. It would be advisable for the first paragraph to present the findings of your study, to later compare your findings with those of other studies (done). Limitations and future lines remain scarce.
  • I still think the conclusions should be more conservative. I think it would be advisable to include “in our study", "preliminary results"... or something similar.
  • Here are my biggest concerns:
    • This comment was not attended, and I think it is an important question.
      • Authors present the differences between boys and girls in the results, but this aspect is not accurately reflected in the previous sections of the paper. Please review this aspect and indicate why gender could be an important factor. Why comparing sex and not, for example, age? 
    • I am worried about the poor characterization of the sample. What about the influence of other socio-cultural-educational-economic factors?
